# A Genomic Information Management System for Maintaining Healthy Genomic States and Application of Genomic Big Data in Clinical Research

**DOI:** 10.3390/ijms23115963

**Published:** 2022-05-25

**Authors:** Jeong-An Gim

**Affiliations:** Medical Science Research Center, College of Medicine, Korea University Guro Hospital, Seoul 08308, Korea; vitastar@korea.ac.kr; Tel.: +82+2-2626-2362

**Keywords:** aging, genomic information, health management, healthy aging, personalized medicine

## Abstract

Improvements in next-generation sequencing (NGS) technology and computer systems have enabled personalized therapies based on genomic information. Recently, health management strategies using genomics and big data have been developed for application in medicine and public health science. In this review, I first discuss the development of a genomic information management system (GIMS) to maintain a highly detailed health record and detect diseases by collecting the genomic information of one individual over time. Maintaining a health record and detecting abnormal genomic states are important; thus, the development of a GIMS is necessary. Based on the current research status, open public data, and databases, I discuss the possibility of a GIMS for clinical use. I also discuss how the analysis of genomic information as big data can be applied for clinical and research purposes. Tremendous volumes of genomic information are being generated, and the development of methods for the collection, cleansing, storing, indexing, and serving must progress under legal regulation. Genetic information is a type of personal information and is covered under privacy protection; here, I examine the regulations on the use of genetic information in different countries. This review provides useful insights for scientists and clinicians who wish to use genomic information for healthy aging and personalized medicine.

## 1. Introduction

“What gets measured gets managed.”—Peter Drucker

In the last decade, improvements in next-generation sequencing (NGS) technology, analysis algorithms, and computer systems have enabled precision or personalized therapy based on genomic information, such as genotype, gene expression, and DNA methylation patterns [1,2,3]. Recently, the development of health management strategies using genomics and big data in medical and public health sciences has come to light [3,4,5]. However, the complex traits of genomic information pose a big challenge for scientists and clinicians in interpreting the patient’s genomic patterns, as well as in optimizing precision or personalized medicine. Human cells reflect diverse intrinsic physiological or pathological state and extrinsic stimuli to maintain homeostasis, and the results of adaptations are enclaved to the landscape of genomics, transcriptomics, epigenomics, proteomics, and metabolomics [6,7]. The following seven points are necessary when discussing the management of genomic big data. First is a sophisticated genomic information-based database to obtain clinical evidence that reflects the pathological conditions. Second is the elucidation of clinical evidence to differentiate between healthy and diseased states. Third is obtaining genomic and clinical information that reflects the age, gender, and health status of each individual. Fourth is the development of a genomic and health information system to quickly and precisely support research hypotheses and clinical decisions. Fifth is insights for use in research and clinical practice from public genome databases, such as the National Center for Biotechnology Information (NCBI), Gene Expression Omnibus (GEO), Sequence Read Archive (SRA), and The Cancer Genome Atlas (TCGA). Sixth is genomic information as big data should be well stored and indexed. Last is that genomic information regulations from the perspective of personal information protection should be balanced between individual privacy and public interest.

Herein, I review the current understanding of these seven points in two parts. In the first part, I discuss genomic information dependent on age and health status and the possibility of developing an information system for determining and maintaining a healthy status. In the second part, I discuss how to extract insights from genomic big data and apply them to research and clinical practice. In addition, I discuss the latest technologies and regulatory science on the storage, indexing, and regulation of genomic information.

## 2. Part 1: Genomic Information Management for Individuals

“Information is the oil of the 21st century, and analytics is the combustion engine.”—Peter Sondergaard

### 2.1. Is It Currently Possible to Develop a Health Management System Based on Genomic Information?

As shown in Figure 1A, generated and collected clinical information undergoes a process. Thus, it is possible to maintain a healthy individual’s health status and use it as evidence for the basis of disease diagnosis and patient prognosis prediction. Association studies between clinical and genomic information have been performed; strategies related to the privacy and security of personalized information need to be suggested. In addition, a standardized clinical data warehouse (CDW) should be established to enable the proper classification and indexing of data. Subsequently, data quality control (QC) and management should be conducted [8]. Currently, standards for clinical and genomic databases (DBs) should be drafted for an ideal maintenance strategy for the established CDW. Through data mining, disease factors can be discovered from the integrated DB, and a predictive model can be constructed [9]. Many methodologies or models that can discover insights from big data have been proposed. Researchers and DB maintainers should select an appropriate model and present an optimal insight discovery strategy for the DB [10,11]. The data obtained through this process should be standardized and shared so that they can be utilized by other research groups. The DB can be integrated or pattern analysis can be performed using artificial intelligence, and a clinical trial can be conducted based on the established model [12,13]. Through this process, it is possible to enable accurate diagnosis and prognosis and to suggest appropriate strategies for maintaining health (Figure 1B). Big data for performing the above process are composed of clinical and genomic information. These big data are still being collected and require lifelogs and personalized information to be collected in the future; therefore, an appropriate storage and indexing strategy is required to compile these data (Figure 1C) [14]. Ultimately, based on the genomic, lifelog, and clinical information, it is possible to propose strategies for treatment intervention, lifestyle modification, and health maintenance by appropriately classifying each individual (Figure 1D).

High-throughput sequencing results obtained using NGS technology have become a crucial part of clinical evidence, and the gaps between sequencing results and conventional clinical tests continue to narrow. Genotype data, such as single-nucleotide variation (SNV) and copy number variation (CNV) data, are used in clinical decisions such as disease diagnosis, prognosis, treatment determination, and drug dosage calculation [15,16]. To apply NGS-based SNV and CNV data to clinical decisions, many studies were performed to reduce side effects and better clinical effects to patients [17,18,19]. Gene expression and DNA methylation patterns are relatively inconsistent and difficult to apply in clinical settings. Due to the complexity of NGS data processing, clinicians find it difficult to interpret the NGS results of patients directly. Bioinformatics is required to extract insights from NGS data. Thus, NGS data interpretation systems have been developed to enable the systemic understanding of genomic and clinical data [20,21]. Ultimately, the Welfare Genome Project (WGP) provided genomic health reports, as well as awareness of how genome projects can benefit health and lifestyle management [22]. The WGP contributed the general understanding of genomics, NGS technologies, bioinformatics, and bioethics regulation. The WGP framework can elicit a new personalized healthcare system based on the systemic understanding of genomic and clinical data [20,22].

In the second and third subsections of this section, I discuss studies on genomic information corresponding to a healthy state and the reasons for obtaining genomic information throughout life. Based on previous studies and accumulated data, it is necessary to continue to secure the basis for finding genetic information related to diseases.

### 2.2. Analyzing Health Status through Genomic Information

Several studies or algorithms have been developed to predict a healthy state based on genomic information [23]. In this review, I discuss only genomic information such as gene expression and DNA methylation patterns, which are altered in response to internal or external stimuli [24,25,26]. As an experimental design for comparing healthy and diseased states, with healthy participants considered as controls [27], a comparative analysis can be performed on age- and sex-matched healthy controls. Blood is the best sample to obtain omics data. It is almost impossible to obtain tissue from a healthy person unless it is donated tissue or normal tissue adjacent to the diseased tissue. Urine omics data do not provide metabolite information [28], and stool omics data do not provide metagenomic information and are not reliable for clinical decision-making [29].

Many studies have investigated what encompasses a healthy genomic state and suggest that it is an optimal state that can be achieved using drugs, healthy foods, nutritional combinations, and exercise [24,30,31]. It is known that a nutritionally balanced diet and regular physical activity affect DNA methylation in a healthy state, and studies have shown that dietary interventions and changes in physical activity can affect DNA methylation patterns [32,33]. In the Make Better Choice 2 (MBC2) study, three 12-week interventions were conducted in adults aged 18–65 years to reduce sedentary time while increasing exercise and fruit and vegetable intake for nine months. Differentially methylated regions (DMRs) in genes related to cell cycle regulation and carcinogenesis were discovered between the intervention and control groups [34]. A study conducted between 2009 and 2010 showed higher LINE-1 and IL-6 promoter methylation with lower C-reactive protein levels and white blood cells in people commuting by public transport (*n* = 101) compared with those in people commuting by car (*n* = 79) [35]. In a study of 161 participants aged 45–75 years, the overall genomic DNA methylation level was significantly higher in those who performed 30 min of physical activity per day than in those who performed less than 10 min of physical activity per day. Physical activity was measured using an accelerometer, and DNA methylation levels were measured using the MethyLight assay [26]. A study investigating LINE-1 and IL-6 promoter DNA methylation status in 165 cancer-free patients aged 18–78 years revealed that among the factors related to age, sex, race, body mass index, diet, and physical activity, only folic acid intake was associated with LINE-1 methylation [36]. Blood-derived DNA methylation analysis of 1016 people over 70 years of age in Sweden divided into four groups according to exercise intensity based on a questionnaire revealed a significant negative association between the locomotor active group and total methylation [24]. In the peripheral blood DNA of the exercise group (*n* = 230) compared to that of the control group (*n* = 153), significantly higher methylation of the ASC gene responsible for IL-1β and IL-18 secretion was observed. Methylation of the ASC gene decreases significantly with age, indicating an age-dependent increase in ASC expression [37].

To date, I have conducted DNA methylation studies related to lifestyle habits such as diet and exercise. Studies in healthy people and comparative studies before and after lifestyle interventions or in randomized groups may facilitate the determination of genomic information reflecting healthy status. It will also facilitate disease diagnosis and prognosis prediction. Subsequent lifestyle interventions may be helpful for a good prognosis and the systematic management of chronic diseases.

### 2.3. Why Should Genomic Information Be Obtained over a Lifetime?

The epigenetic clock measures the degree of accumulation of methyl groups in DNA molecules based on DNA methylation levels to determine an individual’s age. The Hannum epigenetic clock is composed of 71 markers in DNA from blood [38], and the Horvath epigenetic clock measures methylation rates in various tissues using public DNA methylation data [39]. Besides models that predict age by obtaining methylation results from blood or tissue-derived DNA, the PhenoAge or GrimAge clock that uses current age and smoking as additional input data has been proposed [40,41]. The underlying algorithms for PhenoAge and DNAge services are available in the USA. PhenoAge was developed by analyzing data from 9926 people in the National Health and Nutrition Survey III (NHANES III) using machine learning. Subjects in NHANES III were followed up for up to 23 years to determine whether they died or developed cardiovascular disease, cancer, dementia, diabetes, and more. The PhenoAge model provides biological age using current age and nine blood-derived clinical laboratory data, and DNAge predicts chronic disease and mortality risk by comparing the actual age of participants with their DNA-methylation-based age [41]. In GrimAge, seven DNA methylation surrogates and smoking (pack-years) information were used to predict a healthy lifespan. This model uses data from numerous participants related to cancer prevalence, fatty liver disease, and visceral diseases [40].

A database that can predict age based on genomic information is also being developed. To date, many studies have been conducted on age prediction based on gene expression or DNA methylation patterns at each stage of health and disease at specific time points, providing reliable information. However, few studies have obtained long-term time-series genomic data. Therefore, it is necessary to continuously obtain the health and DNA methylation status at each stage, such as adolescence, youth, middle age, adulthood, and old age, throughout an individual’s life (Figure 2A) as they may be valuable in identifying factors explaining why some participants stayed healthy (Figure 2B). Factors that explain the maintenance of a healthy status and those that explain deviation from this state should be determined (Figure 2C,D).

In summary, public genomic information that can predict a healthy state is currently available. However, research on age and health status in individuals is insufficient. Using a similar approach, the first analysis using the Korean Genome Epidemiology Study (KoGES) was conducted on the DNA methylation of 50 blood samples and followed up after approximately eight years with DNA methylation analysis results and health status in the blood [27]. However, there is a need for time-series data from normal samples analyzed using different platforms. In the NCBI GEO database, it is possible to obtain data on healthy persons with similar health statuses; however, different analysis platforms are used for different races. It is still difficult to find common elements and build a generalization or health model. Therefore, it is necessary to establish a basis for constructing a model for aging and health status presentation by identifying factors that can explain age and health status in samples such as donated blood during regular health check-ups [42,43].

Over the past 10 years, DNA methylation patterns due to aging have been observed by several research groups studying epigenetics, including the Horvath group. Mainly microarray-based analyses have been performed on various tissues, including the blood. Studies on DNA methylation changes due to aging have been summarized in a previous study [44] and have shown that DNA methylation or gene expression levels, expressed as age and beta values, are all continuous variables. Therefore, the gene with the highest positive or negative correlation can be determined by calculating the correlation coefficient between age and the beta value, age, and gene expression [45]. In addition, linear regression can be used for constructing a model. Regression analysis or correlation analysis has been performed in most studies on the epigenetic clock.

In the healthy state, the normal aging process induces programmed changes in DNA methylation in cells, accumulated evidence on which can be found in scientific papers, models, and databases [38,46]. A model has been developed in which DNA methylation status and current age data can be fed and data on the current health status and the possibility of disease occurrence can be obtained. Will it be possible to predict and maintain a healthy state using these models (Figure 2B)? Will it be possible to suggest measures to turn a diseased state into a healthy state (Figure 2C)? If the above questions are positively answered, the suggestion of lifestyle change for anti-aging, which can maintain a healthy state, or rejuvenation, which leads to a healthy state from an unhealthy state, will be possible (Figure 2D).

The importance of obtaining data on aging can be summarized as follows: First, by securing a healthy control group of age-matched individuals, it would be possible to compare an individual’s health status with a healthy state based on his/her age. Second, changes in physiological phenomena due to aging could be explained. Third, combining the above two points could help develop drugs, healthy foods, nutritional combinations, and exercise habits that can slow or control aging.

### 2.4. Part 1 Subconclusions

Recently, genomic information has been integrated into the health management system, and DNA methylation data related to individual age, health status, and lifestyle have been accumulated. Epigenetic clock research is becoming more sophisticated, and new factors that can explain a healthy state continue to be identified. However, data from the analysis of an individual’s entire lifetime are still lacking. Strategies for better clinical decision-making must be developed by accumulating and analyzing the DNA methylation state and other clinical data corresponding to healthy, disease-improving, and disease-worsening states.

## 3. Part 2: Big Data-Based Genomic Information Management System for Clinical Research

“Data is the new science. Big data holds the answers.”—Pat Gelsinger

### 3.1. Clinical Decisions and Research Using Genomic Big Data

The clinical decision support system (CDSS) is a tool to improve the treatment effect and prognosis of patients by integrating established clinical knowledge and patient information, as shown in Figure 3. At each stage, from diagnosis to follow-up, the CDSS helps clinicians make optimal decisions based on patient-derived information [47,48]. It is believed that utilizing the CDSS under appropriate regulation greatly optimizes patient treatment. To date, the CDSS has been implemented in pharmacology, pharmacogenomics, laboratory medicine, and pathology [49,50,51,52]. Pregnancy status, renal and hepatic function, drug allergy, drug selection, and dosage for specific diagnostic conditions should be collected. Each parameter, such as drug interaction, chronic disease, and kidney and liver function, is being modeled and refined using patient-derived real-world data [53,54,55,56]. Recently, as the cost of genomic analysis has decreased and accuracy has improved, genotyping of the cytochrome P450 gene, which encodes a protein that metabolizes drugs with narrow therapeutic windows (e.g., tacrolimus and warfarin), has been included in CDSS [50]. Based on this, evidence-based drugs and doses will be determined for individual genotypes.

As of the first half of 2022, several clinical features as input data have been obtained and can be used as a basis for clinical decision-making by using an appropriate analytic tool (Figure 3). Clinical data will be continuously collected. Additionally, scientific literature and clinical guides for prospective and retrospective studies of clinical cases or diseases continue to be published. When these external data are published on the web, they are organized using a web crawler and collected as data [57,58]. The collected data are sufficient in volume and clinical evidence. However, the CDSS in actual hospitals does not provide an active warning, notification, summary, or search system based on the collected data, and several challenges must be resolved.

The collected data are big data, which have disparate and dispersed features. For example, patient information mainly consists of a matrix extracted in the .tsv format and medical images. In the case of genomic information, it is essentially possible to express it in .tsv format, but data generated on various platforms are disadvantageous in that they are incompatible with each other. The two keywords required for this are “standardized” and “structure” for the collected data [59]. The collected data should be described using the term “standardized” and analyzed by the standardized method [60]. In addition, standards must be followed for data generated in the past, being generated in the present, and to be generated in the future. The data collected should be structured to better describe the patient’s symptoms or characteristics. Machine learning and deep learning should be used in this regard. R- or Python-based analysis systems can provide visualization and clinical insights into the data collected by machine learning [61,62].

### 3.2. Storing and Indexing Genomic Information

New insights can be obtained by appropriately integrating and reanalyzing public omics data from NCBI GEO, NCBI SRA, TCGA, and other omics databases (Figure 4A). In this section, I discuss specific strategies for GIMS involving the collection, indexing, classification, and security of genomic information from the aforementioned genomic databases.

Big data on human disease genomics continue to grow not only in number, but also in volume, and the indexing, storage, processing, accessing, and curation of these data have emerged as important challenges. Therefore, it is necessary to use a cloud platform to store and access genomic data, blockchain technology to ensure data security, and sophisticated algorithms for processing, curating, and annotating genomic data for clinical use [63,64]. The goal is to extract valuable insights from raw genomic datasets of TCGA, TCIA, and NCBI GEO as primary data that can be used in clinical practice [65,66,67]. Thus, additional processing, such as data classification, indexing, curation, annotation, and user-friendly access systems, is needed.

However, there are many limitations and challenges associated with using this secondary data-processing database. First, many databases have been created for one use, and many have not been updated since they were created several years ago. Additionally, some databases are inaccessible. Second, there are no commonly agreed protocols for the secondary processing of data. Therefore, each database accepts different types of input queries and presents different output types. Users must learn how to use each database. Third, there are many examples of using the information obtained from the output in research, but little is known of its practical use in clinical practice. Verification studies based on secondary data processing databases and clinical trials targeting patients are required.

In NCBI GEO, the study, analysis platform, and accession numbers for each subject are assigned separately. The accession number for the study starts with “GSE”; the platform starts with “GPL”; the subject starts with “GSM”. ArrayExpress, a database with a similar concept, also annotates and indexes omics data similarly [68,69,70]. Therefore, it is possible to filter using relevant keywords and access the required dataset from the platform according to the research topic. In prospective studies, such as time-series analysis and follow-up, it is recommended to use the same analysis platform as much as possible.

NCBI SRA provides fastq files containing NGS analysis raw data. It is difficult to interpret data in the fastq format for direct clinical use. The fastq format enables diverse data processing using different parameters according to the purpose of the study, as well as offers potential insights to be unveiled by new research groups [71].

The cancer genome database, TCGA, provides raw data on somatic mutations in .vcf and .maf formats. In addition, it provides the gene expression data determined using RNA-seq for 56,000 genes along with their ensemble accession numbers. DNA methylation analysis results are procured from Illumina 27 k and 450 k analysis results. All omics data and clinical data of TCGA are public, except for some controlled samples, and datasets can be received in the form of R data frames through the GDCquery function of TCGAbiolink, which is an R package. The parameters used here are “project” (e.g., TCGA-BRCA) for a total of 33 carcinomas, “data category” (e.g., simple nucleotide variation) corresponding to the omics type, “data type”, which presents detailed analysis and filtering conditions (e.g., annotated somatic mutation), and “workflow type” (e.g., MutTect2 annotation) [72,73].

In Korea, omics data produced with public funding support are deposited in the Clinical & Omics Data Archive (CODA) for medical and human research and the National Agricultural Biotechnology Information Center (NABIC) for animal and plant research. The deposited data can be accessed and downloaded only by authorized researchers after receiving approval from CODA-specific applications of data for a specific period. Although NABIC is not a human database, it has implications as it provides a bioinformation framework that supports research based on the nucleotide sequences of various crops, livestock, and microorganisms [74]. Based on genome sequences, genome browsers have been developed, and comparative genome analysis between different species can be obtained and visualized. Through the comparison, integration, and visualization of genomic data, genome browsers can be viewed as a database that has implications for the prevalence of intergroup and preventive medicine approaches.

Proper indexing of omics data is required to enable researchers to find the data they need quickly and accurately and for administrators to easily manage and aggregate accumulated data to identify and solve problems quickly. In the commonly known NGS analysis process, the alignment result is saved as a .bam file after generating a fastq file. The expression or DNA methylation levels, which are continuous variables, is arranged in a matrix such as in a .tsv file. For categorical variables, variants, and copy number variants, a column containing genomic region information and a column comparing the sequence obtained with the reference genome sequence should exist. TCGA presents data in .vcf, which is a well-known format, and .maf is used for displaying variants [73,75]. An ID system for the participants according to each analysis stage, disease, age, and gender is required to enable the identification of the features of each sample through the assigned ID. According to the hierarchical classification strategy of the TCGA dataset discussed above, only by setting parameters in the GDCquery function, researchers can filter the omics data analyzed under appropriate conditions of the carcinoma under study. Through this strategy, it is possible to secure prior data on the clinical decisions of clinicians, enabling fast and accurate clinical decision-making [76].

Sensitive patient-derived data must be prevented from being accessed, processed, and transferred by unauthorized persons [1,4]. By default, all publicly available omics data are anonymized, making it impossible to identify individuals; attempts to identify them are prohibited in most countries [77]. Attempts to steal, forge, or falsify data from the outside must be prevented by the data manager. The database should implement identification and authentication systems for data use, and only authorized individuals should be able to collect, process, transfer, store, and access data [1,78]. For this purpose, the integrity of clinical and omics data should be maintained, and there should be an intrusion detection plan. In addition, there must be countermeasures for disaster recovery, and a protocol to determine at what level and who should be held responsible in case of a security violation [79].

For patient-derived genomic and clinical data, periodic updates and back-up are essential for proper treatment or patient management in hospital. In GIMS, the periodic update and back-up of patient-derived data could be utilized by technological methods applied to medical big data, such as medical images [80,81]. The methods provide the security, privacy, and back-up architecture for patient-derived data. The linkage with electronic health records (EHRs) in hospitals and standardization of EHRs are also considerations, and the use of an external cloud system such as the Amazon Web Services (AWS) cloud is also a possible alternative [82,83].

In this review, GIMS is aimed a global approach. However, two region-specific issues are presented. First, race-specific genomic data should be based on GIMS. In the WGP study, genome sequencing projects performed in various countries and the purpose of each project were presented [22]. More than 10,000 genome sequencing data will be obtained from each project such as the 100,000 Genome projects in England, FinGen in Finland, the 100,000 Genomes Project in China, the Genomics Thailand Initiative, and the Saudi Genome Program. The genome data for each race will provide clinical evidence for GIMS by region. In the next section, the general regulation of genomic information will be discussed, and the details of each country’s regulations were discussed in the cited papers.

### 3.3. Regulation of Genomic Information: In Terms of Personal Information and Privacy Protection

Technological development and an increased understanding of sequencing have led to issues in regulating the availability and use of personalized sequencing data. Every country has a strategy for the disclosure and regulation of personal genomic data [77]. Excessive disclosure of data can lead to an invasion of privacy, and excessive regulations make it difficult for researchers to use genomic data [79,84]. Therefore, balancing the regulation and disclosure of data is an ongoing discussion (Figure 4B).

Large-scale and diverse clinical and omics data can optimize precision medicine. Therefore, it is important to collect a large amount of data; however, the larger the data, the more difficult it is to manage and secure. The leakage of personal information affects individuals’ lives through bullying, high insurance rates, and unemployment due to medical history [79,85]. Therefore, security, privacy, and trust in managing patient information are essential. In addition, government legislative and ethical committees warrant the security and privacy of medical data. Besides, individuals expect security, privacy, and trust with their data. If this is not ensured, people may stop providing their data to precision healthcare systems. Consequently, the effectiveness of precision health diminishes as the public becomes the target beneficiary of the system. It is important to find the best practices and techniques for leveraging health data in light of precision health data security, privacy, ethical, and regulatory requirements [77,84]. One of the best practices was a nation-wide survey on the citizen participation cohort program for the Resource Collection Project for Precision Medicine Research (RCP-PMR) before the project proceeded [86]. In a survey by Kim et al., it was found that Koreans were more likely to need a national precision medicine cohort program, to participate in the project, and to provide samples, compared to Americans and Japanese. However, participants with negative attitudes were concerned about privacy violations and did not consent to data sharing with researchers other than government researchers. To overcome this, it is necessary to communicate how data sharing helps medical care, and a discussion that includes the opinions of various stakeholders such as civic groups, patient groups, and researchers is necessary [86].

Various countries have a regulatory system for clinical and omics data and require close observation. In general, written consent provided by each participant is essential when using his/her health information. In addition, the confidentiality of data should be protected administratively, physically, and technically. If the requirements of regulatory systems are violated, they must be notified individually [77,79,84].

Therefore, it is necessary to ease restrictions on the use and disclosure of anonymized clinical and omics data. However, considering that these data still carry the risk of re-identification and civil and monetary liability, criminal penalties for unauthorized re-identification should be imposed [77]. Additionally, the process of the disclosure, use, and utilization of de-identified data should be informed, and the method for de-identification should be provided [79]. When the participant providing data has suffered damage intentionally or negligently, it is necessary to establish a process for damage estimation and relief and to enact legislation in their support. The institutional review board (IRB) in each institution understands the characteristics of clinic and omics data well, and if the requirements related to the use of data in this section are satisfied, the data must be allowed to be processed and disclosed in the form of scientific papers for secondary reprocessing.

### 3.4. Part 2 Subconclusions

In recent years, the amount of patient-derived data available for oncology or chronic disease research has increased rapidly, and the following three strategies are suggested for their optimization: First, clinical and genomic data for clinical use must be continuously accumulated, and regulatory bodies must provide permission for new drug development and companion diagnosis based on rational evidence and data. Second, appropriate storage, indexing, and security strategies should be implemented for both clinical and genomic data management. Finally, it is important to balance public interest and the privacy and security of personal information, and the IRB should permit the processing of data based on reasonable grounds.

## 4. Future Perspective

The future of health management systems will involve a GIMS composed of genomic information. The GIMS may enable estimating the risk for disease, borderline, and healthy state using a well-established model that uses the genomic, clinical, laboratory, and lifestyle data of a patient or normal person (control) as input data (Figure 4C). With the current information, hardware, analysis libraries, and management systems can be built for data processing. However, the relationship between genomic information and treatment strategy should be further established, and the regulatory body should establish ideal review or approval criteria for GIMS approval. Through this, it would be possible to systematically manage an individual’s health.

## Figures and Tables

**Figure 1 ijms-23-05963-f001:**
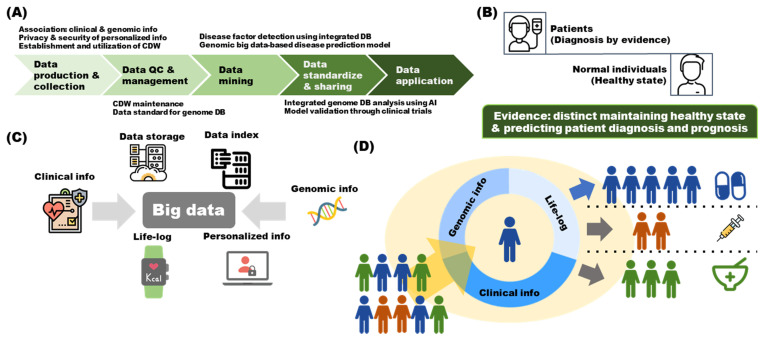
Overview of the genomic-information-based health management system. (**A**) The five processes required for the utilization of clinical and genomic information. (**B**) Clinical and genomic information should provide evidence to enable diagnosis or prognosis and to predict healthy status in normal individuals. (**C**) Along with clinical and genomic information, big data from various sources can be used for clinical decision-making through appropriate storage and indexing. (**D**) The final goal is to suggest an appropriate strategy for health management in healthy conditions and to provide customized treatment in diseased conditions, based on all data obtained for each individual. AI, artificial intelligence; CDM, common data model; CDW, clinical data warehouse; DB, database; info, information; QC, quality control.

**Figure 2 ijms-23-05963-f002:**
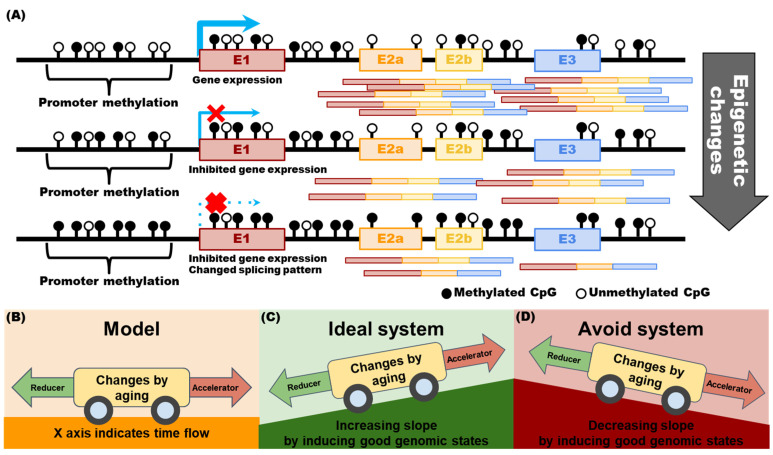
Changes in gene expression and splicing patterns following changes in DNA methylation by aging or disease progression explained via a car model. (**A**) When CpG sites located in the upstream region of the transcription start site are hypermethylated, gene expression decreases. When the CTCF binding site becomes hypermethylated, exon skipping occurs. In this epigenetic pattern change model, as gene hypermethylation increases, the number of transcripts and skipping phenomenon of exon 2b decrease. (**B**) Humans age and undergo changes in DNA methylation patterns over time. Factors that accelerate or reduce aging have been discovered. (**C**) A car model proposing the use of the genomic information management system presented in this review as an ideal system for delaying aging. (**D**) Maintaining detrimental lifestyle habits, irregular and inaccurate health screenings, and improper restrictions by the government on the use of genomic information can accelerate aging.

**Figure 3 ijms-23-05963-f003:**
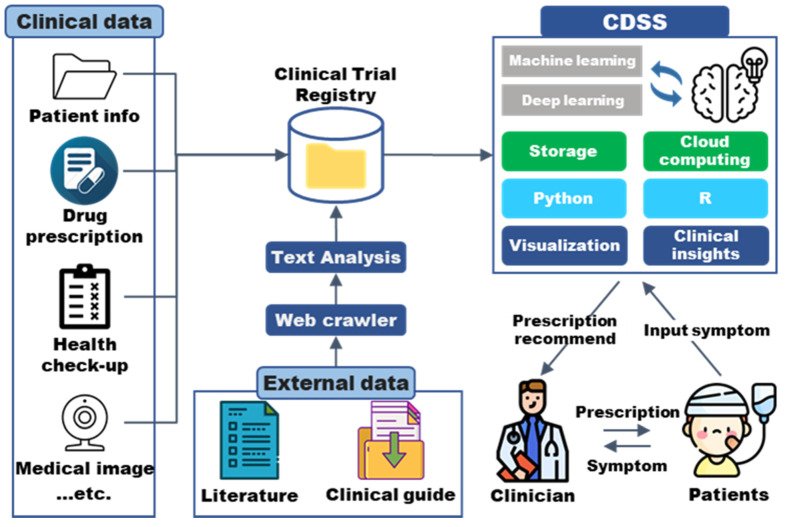
Integrated strategies of the clinical decision support system (CDSS). Clinical data consist of patient information, drug prescription, data from health check-ups, and medical images. These data should be cleaned and indexed in a format commonly used in the field. In addition, as external data, scientific papers and clinical guidelines can be deposited in storage or cloud computing systems using web crawler or text mining tools. Machine learning can be used as an algorithm for CDSS; the languages mainly used are Python and R. Machine learning aims to visualize this appropriately and provide patient-specific clinical insights to clinicians. Using the CDSS developed as the model, researchers can obtain clues to the development of a genomic information management system (GIMS).

**Figure 4 ijms-23-05963-f004:**
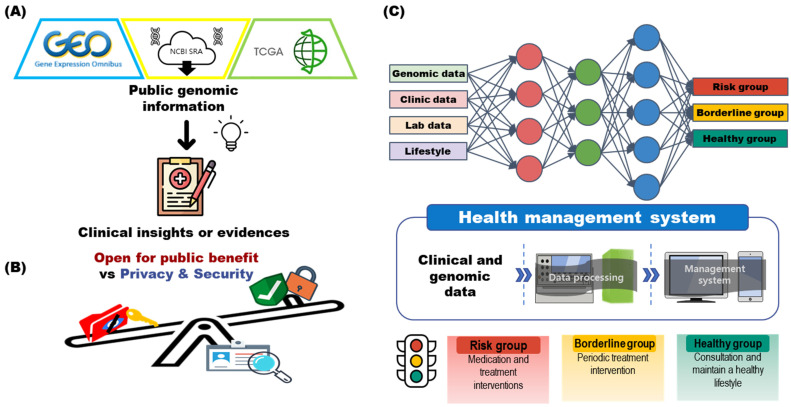
Example of an ideal genomic information management system (GIMS). (**A**) Many public omics data are deposited in NCBI GEO, NCBI SRA, and TCGA. These can be used as reference data for analyzing omics data to be produced in the future and can help discover clinical insights or evidence. (**B**) When using omics data, a compromise must be found between the disclosure of information for public benefit and maintaining the privacy and security of patients and participants. (**C**) In the GIMS, individual genomic, clinical, laboratory data, and questionnaire-based diet and exercise information can be classified into risk, borderline, and healthy groups using machine learning. This can be generalized and used as a health management system and can facilitate the assessment of the need for treatment intervention, as well as aid the provision of lifestyle-related suggestions for maintaining a healthy state.

## Data Availability

Not applicable.

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
