# Peer review of "A Genomic Information Management System for Maintaining Healthy Genomic States and Application of Genomic Big Data in Clinical Research"

_ijms, 2022, doi:10.3390/ijms23115963_

Round 1

Reviewer 1 Report

Overall, the manuscript entitled “A genomic information management system for maintaining healthy genomic states and application of genomic big data in clinical research” is a necessary review about standardization of genomic data applied in clinical environment. After two corrections, I think the manuscript will be optimum to publish in this journal.

Specific comments:
- The whole manuscript is well written with good concision and clarity. Tables and figures are very appropiate.
- Please, correct the incorrect written name Horvath in row 197.
- References: In opinion of this reviewer, there is no relation between the paragraph "Blood-derived DNA methylation analysis of 1,016 people over 70 years of age in Sweden divided into four groups according to exercise intensity based on a questionnaire revealed a significant negative association between the locomotor active group and total methylation" (rows 132-135) and the reference linked ([20]). An explanation of which part of the referenced article has been extracted to justify that paragraph is very appreciated.

Author Response

Overall, the manuscript entitled “A genomic information management system for maintaining healthy genomic states and application of genomic big data in clinical research” is a necessary review about standardization of genomic data applied in clinical environment. After two corrections, I think the manuscript will be optimum to publish in this journal.

→ Thank you very much for your review for the development of this paper. Two points were correlated.

Specific comments:

- The whole manuscript is well written with good concision and clarity. Tables and figures are very appropiate.

- Please, correct the incorrect written name Horvath in row 197.

→ Corrected to be the correct spell for “Horvath” [Line 204].

- References: In opinion of this reviewer, there is no relation between the paragraph "Blood-derived DNA methylation analysis of 1,016 people over 70 years of age in Sweden divided into four groups according to exercise intensity based on a questionnaire revealed a significant negative association between the locomotor active group and total methylation" (rows 132-135) and the reference linked ([20]). An explanation of which part of the referenced article has been extracted to justify that paragraph is very appreciated.

→ There was a problem in transferring the citation information to the Endnote program. The author meticulously checked whether the rest of the parts were also properly cited, and the correct study (Luttropp et al., 2013) was cited. As a result of checking the rest of the parts thoroughly, all 86 citations including papers added after revision were accurately expressed. However, one typo error found while checking this, "MethylLight" → "MethyLight" was corrected [Lines 135-136].

Reviewer 2 Report

This comprehensive review is an attempt to provide useful insights on managing genomic information for application in clinical research. It is a very interesting review as it deals a very important aspect of clinical research i.e. building a relationship between  genomic information and treatment strategy. However there are several points that need to be addressed in order to make this review more insightful.

1. Throughout the review and under different subsections, author explains the current challenges and limitations in dealing with genomic information. However it is very important to shed light on the probable solutions for these challenges/limitations. Technical details e.g. details on DNA methylation and how it effects gene expression and other such basic technical details should be reduced and the focus should be on discussing probable solutions for the present challenges.

2. It is not clear if author is trying to discuss a global approach for establishing the genomic information system or is it region specific? If it is global, it is important to have region specific genomic data. For example, there are several diseases that occur in only certain regions of the world. Please comment.

3. GIMS require frequent updates for proper treatment in the clinic. How can this upgradation be taken care of? Is there any automatized system that would upgrade on a weekly/monthly basis?

4. GIMS will have an enormous amount of valuable data and it is very important to have a backup for this data. Please comment.

5. Are there already established GIMS? If yes, author should compare those and highlight the utility and advantages of the GIMS present till date.

6. Author has discussed only about gene expression data and DNA methylation data as genomic information. There are several other genomic data which are very important and useful for clinical research e.g. SNP, copy number variations. In addition data involving various molecular pathways and  drug repurposing data would be very helpful. Please include a discussion with the above points and other genomic information data that might be useful.

7. As mentioned on Line 279:"but data generated on various platforms are disadvantageous in that they are incompatible with each other". Data incompatibility is a concern when using GIMS. Please discuss a probable solution for this.

8. Line 391 mentions : "It is important to find the best practices and techniques for leveraging health data in the light of precision health data security, privacy, ethical, and regulatory requirements". Please mention the best practices briefly.

9. Please check for typos throughout the manuscript. For example, Line162: "for PhenoAge, PhenoAge, and DNAge , reputation". Please replace the repetitive word.

Author Response

This comprehensive review is an attempt to provide useful insights on managing genomic information for application in clinical research. It is a very interesting review as it deals a very important aspect of clinical research i.e. building a relationship between  genomic information and treatment strategy. However there are several points that need to be addressed in order to make this review more insightful.

  1. Throughout the review and under different subsections, author explains the current challenges and limitations in dealing with genomic information. However it is very important to shed light on the probable solutions for these challenges/limitations. Technical details e.g. details on DNA methylation and how it effects gene expression and other such basic technical details should be reduced and the focus should be on discussing probable solutions for the present challenges.

→ Thank you very much for the constructive suggestions. The author is very interested in probable solutions, and has tried to put possible solutions into this manuscript. For basic technical details, if you provide a specific part that needs to be reduced, it will be reflected in the next round. For the points below suggested by the reviewer, the author agreed with all of them, and did my best to revise them. Please check the corrections and improvements. 

  1. It is not clear if author is trying to discuss a global approach for establishing the genomic information system or is it region specific? If it is global, it is important to have region specific genomic data. For example, there are several diseases that occur in only certain regions of the world. Please comment.

→ The reviewer gave an accurate point. Although GIMS aims to take a global approach, there are two region (or nation)-specific issues. One for ethics-specific genomic data and the other for country-specific regulations. Each country-led genome project was presented in the form of a supplementary table in Jeon et al., 2021, and some were described in this paper (> 10,000 participants) [Lines 391-397]. When connecting from the fifth section to the sixth section, I added the descriptions of each country's regulation. Discussion on regulations in each country is beyond the scope of this review, and has already been dealt with in a cited paper [Lines 398-399]. Additionally, in the sixth section, the Resource Collection Project for Precision Medicine Research (RCP-PMR) in Korea was added [Lines 423-432].

  1. GIMS require frequent updates for proper treatment in the clinic. How can this upgradation be taken care of? Is there any automatized system that would upgrade on a weekly/monthly basis?
  2. GIMS will have an enormous amount of valuable data and it is very important to have a backup for this data. Please comment.

→ Reviewer made two valuable points, which are related to the fifth section. I presented the contents of collection, indexing, classification, and security for genomic information. Contents related to upgrade and back-up were discussed and related papers were cited [Lines 383-390].

  1. Are there already established GIMS? If yes, author should compare those and highlight the utility and advantages of the GIMS present till date.

→ To the best of my knowledge so far, an ideal GIMS has not been implemented. However, the most similar project to implement this is “Welfare Genome Project (WGP)” in Korea. This improved the citizen's understanding of genomics, NGS technologies, and bioethics, and suggested the importance of health management. I added discussions in section 1 for WGP [Lines 98-103].

  1. Author has discussed only about gene expression data and DNA methylation data as genomic information. There are several other genomic data which are very important and useful for clinical research e.g. SNP, copy number variations. In addition data involving various molecular pathways and  drug repurposing data would be very helpful. Please include a discussion with the above points and other genomic information data that might be useful.

→ I fully agree with the reviewer's opinion. In fact, compared to gene expression and DNA methylation information, data such as SNP and CNV are clinically more important, and a lot of clinical evidence is presented. However, if all of them are discussed, the scope of the paper becomes too wide. Therefore, in this review, I mainly discussed gene expression and DNA methylation, which show differences according to lifestyle in longitudinal analysis. However, by actively reflecting the reviewer's opinion, a study with a similar scope to this review in relation to SNP and CNV was mentioned and cited [Lines 89-90, and 92-93].

  1. As mentioned on Line 279:"but data generated on various platforms are disadvantageous in that they are incompatible with each other". Data incompatibility is a concern when using GIMS. Please discuss a probable solution for this.

→ Data incompatibility is a very important concern when using GIMS, and I provided two keywords: “standardized” and “structure”. I tried to add a probable solution for data incompatibility, and I added one citation to discuss the concern (Dash et al., 2019) [Line 289].

  1. Line 391 mentions : "It is important to find the best practices and techniques for leveraging health data in the light of precision health data security, privacy, ethical, and regulatory requirements". Please mention the best practices briefly.

→ I added one survey, which provides the possibility of leveraging health data in Korea (Kim et al., 2020) [Lines 423-432].

  1. Please check for typos throughout the manuscript. For example, Line162: "for PhenoAge, PhenoAge, and DNAge , reputation". Please replace the repetitive word.

→ Modified [Line 169].

Round 2

Reviewer 2 Report

Author has satisfactorily responded to all comments.